# Clinical Application of Oncolytic Viruses: A Systematic Review

**DOI:** 10.3390/ijms21207505

**Published:** 2020-10-12

**Authors:** Mary Cook, Aman Chauhan

**Affiliations:** 1Department of Internal Medicine, Marlene and Stewart Greenebaum Comprehensive Cancer Center, University of Maryland Medical Center, 22 S. Greene Street, Baltimore, MD 21201, USA; Maryacook@umm.edu; 2Department of Internal Medicine-Medical Oncology, University of Kentucky, Lexington, KY 40536, USA; 3Markey Cancer Center, University of Kentucky, 800 Rose Street, Lexington, KY 40536, USA

**Keywords:** immunotherapy, clinical trials, oncolytic viruses

## Abstract

Leveraging the immune system to thwart cancer is not a novel strategy and has been explored via cancer vaccines and use of immunomodulators like interferons. However, it was not until the introduction of immune checkpoint inhibitors that we realized the true potential of immunotherapy in combating cancer. Oncolytic viruses are one such immunotherapeutic tool that is currently being explored in cancer therapeutics. We present the most comprehensive systematic review of all oncolytic viruses in Phase 1, 2, and 3 clinical trials published to date. We performed a systematic review of all published clinical trials indexed in PubMed that utilized oncolytic viruses. Trials were reviewed for type of oncolytic virus used, method of administration, study design, disease type, primary outcome, and relevant adverse effects. A total of 120 trials were found; 86 trials were available for our review. Included were 60 phase I trials, five phase I/II combination trials, 19 phase II trials, and two phase III clinical trials. Oncolytic viruses are feverously being evaluated in oncology with over 30 different types of oncolytic viruses being explored either as a single agent or in combination with other antitumor agents. To date, only one oncolytic virus therapy has received an FDA approval but advances in bioengineering techniques and our understanding of immunomodulation to heighten oncolytic virus replication and improve tumor kill raises optimism for its future drug development.

## 1. Introduction

Enhancing the body’s own response to malignant cells through immune stimulation has been a vigorous focus of recent cancer research. Oncolytic viruses (OVs) are one such tool. These viruses are naturally occurring or can be modified to selectively infect and destroy cancer cells. In addition, there is evidence OVs can stimulate the host’s immune response to combat tumors [1]. Multiple viruses are currently under investigation including herpesvirus, adenovirus, poxvirus, picornavirus, and reovirus as possible oncolytic treatments. In 2015 talimogene herparepvec was the first OV approved by the Food and Drug Administration (FDA) for human use in the USA [2] and there are a number of other OVs currently in phase III testing. Here we discuss how OVs have been adapted to destroy cancer cells and summarize the clinical data on OVs currently under investigation.

## 2. Methods

In our systematic review we collected all published clinical trials that utilized oncolytic viruses. Using Pubmed, we first narrowed our search to include “Clinical Trials” and “Humans.” Subsequent search criteria included “oncolytic virus”, “oncolytic virotherapy”, “oncolytic immunotherapy”, and “oncolytic vaccine”. Our initial search returned 120 articles. This preliminary search also captured reviews, proposed study protocols, and opinion pieces, which were excluded (26 in total). Papers not available in full text for review, or with no available English translation were excluded (4). Last, we found a limited number of articles that described different facets of the same trial, and these redundant articles were excluded (4). One unpublished study was identified in clinicaltrials.gov. In summary, we used 86 total trials, including 60 phase I, 5 phase I/II, 19 phase II, and 2 phase III trials for our review. See Figure 1 for a schematic representation of systematic review of OV clinical trial data and exclusion criteria.

## 3. Mechanism of Action

OV therapy relies on a two-part process of selectively infecting tumor cells, and then inducing antitumor activity through release of tumor antigens and immune stimulation (Figure 2). 

## 4. Targeting Cancer Cells

OVs can target and destroy cancer cells by taking advantage of a tumor’s unique cellular activity. Strategies for infection and destruction of the tumor cells vary among OVs. Cancer cells can have altered or entirely absent signaling pathways that normal tissues use to defend against viral infection. 

As an example, protein kinase receptor (PKR) and its associated interferon (IFN) pathway is integral to viral clearance, but is often underexpressed in certain cancers; low IFN expression renders the cancer cells susceptible to viral attack [3]. Another mechanism centers on the increased metabolic activity of tumor cells and rapid replication; for virally infected tumor cells this leads to increased viral replication and rate of lysis compared to normal tissue [4]. This situation enhances viral activity to destroy tumor cells. 

Naturally occurring viruses already contain many strategies for evading host immunity and can take advantage of decreased defenses to target cancer cells. These cells often overexpress receptors or other proteins that the viruses already use you gain entry to normal cells. HSV-1 variants use nectins and herpes virus entry mediator (HVEM), expressed on cancer cell membranes, to gain entry. Some melanomas and carcinomas have increased expression of nectins and HVEM [5]. Measles virus utilize the CD46 receptor, which is often overexpressed in certain cancers [6]. 

Other OV’s are enhanced viruses that have been tailored to increase their affinity for malignant tissues. They are engineered to target proteins overexpressed in cancer cells. One such example is thymidine kinase (TK). A strain of HSV has been modified to favor tumor cells due to their high TK expression [7]. TK is an enzyme used for DNA synthesis and repair during cell replication, is therefore highly expressed on tumor cells. This engineered variant of HSV holds a TK-knock out. The knockout of TK results in preferential infection and destruction of cancerous tissues. 

## 5. Direct Tumor Cell Lysis

After target infection of the tumor cell occurs, lysis of the tumor cell leads to the release of viral particles, cytokines, and other cellular contents, and a secondary response to this lysis ensues. Within this system, there is direct killing of surrounding cells by the release of these cytotoxic elements, which include granzymes and perforins. The cellular contents ATP, uric acid, and heat shock protein are known as danger-associated molecular patterns (DAMPs), which cause local inflammation. DAMPs and cytokines attract natural killer (NK) cells to the surrounding tissue. In addition, viral particles have protein sequences that can stimulate the immune system, known as pathogen associated molecular patterns (PAMPs). 

OVs can also be engineered to deliver suicide genes to enhance direct tumor cell lysis. These genes encode enzymes that can convert prodrug into an active form to destroy tumor cells. ProstAtak (*aglatimagene besadenovec*) is an OV engineered to express HSV TK in infected cells; after injection, valcyclovir is administered that halts DNA replication in the tumor cell.

## 6. Antitumor Activity

OV antitumor activity depends on both direct malignant cell destruction as well as a stimulating systemic antitumor immunity. This immunity is induced when an infected cell is lysed. The viral antigens and cellular components are liberated into the cancer microenvironment. Release of DAMPs, PAMPs, and cytokines lead to maturation of antigen presenting cells (APCs), including the dendritic cells (DC). By these processes, tumor cells that had previously evaded the immune system can now be recognized and targeted by the immune system; destruction pattern recognition receptors (PRRs) on immune cells are stimulated to recognize these PAMPs and DAMPs. PRRs specific to dendritic cells (DC)s within tumors can be directly stimulated by OV infection [8]. The subsequent activation of APCs then recruit CD4+ and CD8+ cells to destroy cells expressing viral antigens on tumors. One such example, in the activation antiviral CD8 T cells by reovirus lead to tumor regression [9]. This recognition of tumor antigens by the immune system is key to tumor destruction at distant sites that were not infected with the OV [10].

HSV and reovirus can prime antitumor immunity by stimulating T cell activity to destroy neighboring tumor cells [11,12]. Another genetically modified variant of HSV, Talimogene herparepvec (T-vec), has the GM-CSF cistron inserted into its genome. This modification enables high production and release of GM-CSF when the tumor cells are destroyed, resulting in enhanced DC recruitment and antigen presentation [13]. 

In addition to targeting the malignant cells themselves, OVs can destroy tumor vasculature. Some OVs are engineered to target angiogenesis, and other viruses inherently target the destruction of tumor vasculature. HSV and vaccinia viruses are one such example. These viruses depend on high expression of fibroblast growth factor and vascular endothelial growth factor (VEGF) for replication to target new tumor vessels [14,15]. The vesicular stomatitis virus (VSV) has been shown to infect blood vessels surrounding a tumor and cause thrombosis of its vessels [16].

## 7. Novel Trends in OV Advancements

Much of the emerging research for OVs focuses on how to amplify their immunogenicity and tumor destruction through insertion or exchange of human genes. These genes can improve cancer cell destruction through improved entry into cells, direct destruction of cancer cells, or recruitment of the patient’s immune system. These emerging viruses are particularly focused on recruitment of the host’s immune system to destroy the tumor.

One such virus in an ongoing phase 1 clinical trial utilizes a strain of HSV-1 to target glioblastomas (NCT02062827) [17]. This OV was modified to express IL-12, which improves the OV’s efficacy in destroying tumor cells and in the surrounding microenvironment. The release of IL-12 with tumor lysis leads to recruitment of dendritic cells, monocytes, and macrophages to the area for increased tumor destruction. The release IL-12 also exhibits an anti-angiogenic effect, which slows the growth of blood vessels for the tumor and decreases tumor growth. 

TILT-123 is another exciting example of an OV equipped with proteins to enhance its anti-tumor activity. TILT-123 is an adenovirus engineered to express TNF-alpha and IL2 (NCT04217473) [18]. While infection with the OV can lead to tumor destruction as previously discussed, the addition of IL2 and TNF-alpha increases the immunogenicity of the therapy. Expression of TNF-alpha and IL-2 attracts T cells to the tumor and also enhances their infiltration into the cancer cells [19]. NCT04217473 is the first clinical trial to evaluate the safety of TILT-123′s in patients with advanced melanoma, enrolling patients in early 2020. 

In addition to studies currently underway combining OVs with simultaneous checkpoint inhibitor therapy administration, viruses are being engineered to produce the antibodies themselves. An HSV-1 virus has recently been developed for use in glioblastoma therapy, which can express PD-1 antibodies [20]. While this construct has only been tested in murine models as of now, it is a promising example of enhancing OV tumor activity through gene insertion. 

## 8. Augmentation of Immune Checkpoint Inhibition. “Making Cold Tumors Hot: Immune Checkpoint Inhibitor–OV Combination Therapy Trials”

Remarkable progress has been made in the development of immunotherapy to treat cancers in the last decade. One type of immunotherapy is immune checkpoint inhibitors (ICBs), which are monoclonal antibodies that block receptor-ligand interactions that negatively regulate both innate and adaptive immunity. Prominent examples of ICBs include those that target cytotoxic T lymphocyte antigen 4 (CTLA-4) and programmed cell death protein 1 (PD-1) [21]. To date, multiple ICBs have been approved by the FDA to treat cancers; for example, ipilimumab (blocks CTLA-4), nivolumab (blocks PD-1), and pembrolizumab (blocks PD-1). 

Combination therapies that use ICBs and oncolytic viruses are attractive. The oncolytic virus can recruit tumor-infiltrating lymphocytes into immune-deficient tumors and trigger the release of soluble tumor antigens, danger signals, and proinflammatory cytokines, which further increase T cell recruitment and promote immune cell activation [10,22]. Viral infection also increases the expression of CTLA-4, PD-1, and other immune checkpoint molecules, which would usually block T cell activation (and therefore antitumor immunity) but also sensitizes tumors to ICBs [23,24]. Non-clinical studies using B16–F10 melanoma demonstrated that localized injection of tumors with oncolytic Newcastle disease virus induced infiltration of tumor-specific CD4+ T cells and CD8+ T cells into both the injected tumor and distant tumors; this action increased the sensitivity of the tumors to systemic CTLA-4 blockade [10]. In another non-clinical model of triple-negative breast cancer, an oncolytic Maraba virus had activity as a neoadjuvant therapeutic, and it sensitized otherwise refractory tumors to ICB [25]. Other oncolytic viruses, such as B18R-deficient vaccinia virus [26] and vesicular stomatitis virus, express a library of melanoma antigens [27] and also showed significant (*p* < 0.05) therapeutic benefit in combination with ICB. 

There is a strong mechanistic rationale for using combination oncolytic virus and ICBs since immune activation in the tumor environment as well as neo antigen production from oncolytic virus can improve efficacy of ICBs. 

Ribas et al. confirmed that OV can potentially improve efficacy of anti PD1 immune checkpoint inhibitors. They evaluated safety and efficacy of combination T- VEC (talimogene laherparepvec) and pembrolizumab in a Phase 1b melanoma clinical trial (*n* = 21) and confirmed 62% objective response rate. The investigators found that OV can favorably alter tumor microenvironment (elevated CD8+ T cells, upregulation of INF-gamma, and increased expression of PD-L1 protein) resulted in improved response to the immune checkpoint inhibitor [28]. 

Improved efficacy was shown in Phase 2 clinical trial of combination oncolytic virus and CTLA-4 targeting ICBs [29]. Chesney et al. evaluated the combination T-VEC and Ipilimumab in a randomized phase II melanoma study (*N* = 198). 39 % patients had objective responses in the combination T-VEC/Ipilimumab arm while only 18% objective response was observed in the single agent ipilimumab arm (95% CI, *P* = 0.002). Of note, the responses were seen in injected lesions but also reported in distant visceral lesions in 52 % patients in the combination arm vs. only 23 % in the single agent ipilimumab arm.

A novel oncolytic virus called Coxsackievirus A21 (CAVATAK) shows synergy when combined with immune checkpoint inhibitors [30]. The CAVATAK virus makes entry to cancer cells via ICAM1, which is often abundant on tumors. A clinical trial evaluating cavatak virus in combination with ipilimumab showed 50% objective responses in melanoma patients. Similarly, a trial evaluating combination cavatak virus with pembrolizumab in advance solid tumors reveled no dose limiting toxicities (DLT) [31]. A phase 1b Keynote 200 clinical trial of combination cavatak + pembrolizumab in advance non-small cell lung cancer and bladder cancer confirmed no DLT; 12.8% grade 3 treatment-related adverse events were noted, however no grade 4 or 5 toxicities were seen [32,33]. Table 1 summarizes combination immune checkpoint and oncolytic virus clinical trials.

## 9. Modes of OV Delivery 

Depending on both the OV and cancer, virus administration can be delivered using a systemic or more targeted approach.

### 9.1. Intralesional Route

A locoregional approach has been used for melanoma, prostate, and gliomas. Intradermal injections of T-vec is employed for melanoma, intracavitary injection for gliomas, and intraperitoneal injection for treatment of ovarian cancer are a few examples. Depending on the location and accessibility of the tumors the virus can be delivered once (i.e., to a glioma cavity during surgery) or rely on repeated injections such as in melanoma [35]. The efficacy of intratumoral injection can extend beyond the lesions injected. Studies report evidence of tumor lysis in satellite lesions not injected with virus [36]. OV delivery can also employ intravenous (IV) release, through either peripheral IV injection, or more targeted, through hepatic artery infusion for liver metastasis [37].

Intralesional injection can be limited by the extracellular matrix that surrounds otherwise accessible tumors. In desmoplastic tissues, the spread of the virus can be limited by excessive fibrin and connective tissues [38]. Agents such as bacterial collagenase have been used to increase the spread OV injection for local tumors such as melanoma, which demonstrated some efficacy [39]. However, collagenase cannot be used systemically, which limits its application. One of many systemic agents under investigation includes the antihypertensive agent losartan, which reportedly decreased collagen deposition, increased blood flow as well as uptake of the cytotoxic agent in pancreatic cancer [40]. 

Vascular injection also has limitations. Tumor vasculature is markedly disorganized and more unstable than normal vessels [41]; this property makes vascular injections less reliable for downstream delivery of the virus to surrounding cancer cells. However, currently there are several avenues under investigation to stabilize these tissues and enhance delivery of cancer immunotherapy; some examples include the use of nitric oxide [42] and VEGF [43].

### 9.2. Intravenous Route

Intra lesional OV administration is currently by far the most commonly used delivery method. However many trials are also evaluating intravenous route of administration [44,45] due to its inherent advantages. Some of the advantages of IV route includes ease of administration, standardization of administered drug dose, potential for multiple and long term administration, and wider application of IV based treatment especially in smaller community clinics [46]. The key detriment to IV administration so far has been development of neutralizing antibodies and clearance of OV [47,48,49]. There are at least eight clinical studies that we can confirm which used IV route for OV administration. 

## 10. Phase I Trials

In our systematic review we have summarized 59 phase I trials investigating 36 OVs. Phase I trials seek to demonstrate safety and dosing. Of those 36 viruses, many did not proceed past this initial phase of investigation, or were altered in some way to improve efficacy. It is not feasible to summarize all Phase I trials here, but a summary of all reviewed trials can be found in Table 2 Here we discuss a few OVs of interest, many of which show promise and have progressed to Phase II investigations. 

## 11. Pexa-Vec

Pexa-Vec/JX-594, or pexastimogene devacirepvec, is an oncolytic pox virus that has been studied across a variety of malignancies, including melanoma, colorectal cancer, and liver metastases. This vaccinia virus was altered for oncolytic activity by deactivation of its thymidine kinase, with the addition of granulocyte–macrophage colony-stimulating factor (GM-CSF). The GM-CSF is added to increase oncolysis and induce adaptive immune response to the tumor cells [107]. A majority of the studies focused on liver cancers, either HCC or metastases to liver. In one study, 14 patients received intratumoral injections of the virus. Grade III hyperbilirubinemia was observed with the highest dosed group, and 4 patients experienced transient thrombocytopenia, as high as grade 3. Under RECIST criteria, 3 patients had a partial response (PR) and 6 had stable disease (SD). Interestingly, a response could be seen not only in the tumors that had been injected, but also in the satellite lesions [108]. Pexa-vec has also been studied using IV administration. In a study of 15 patients with colorectal cancer, IV infusion of Pexa-vec resulted in SD in 67% of patients [94]. Pexa-vec showed promise in phase II clinical trials (see Table 3), but the Phase III trial that combined Pexa-vec with sorafenib in patients with metastatic HCC (Clinical Trial: NCT02562755) was terminated in august 2019 due to lack of effectiveness. 

## 12. Ad5/3 delta 24 (CRAd)

Ad5/3 delta 24 is an oncolytic adenovirus being studied for application across a number of malignancies, and prominently with gynecologic cancers. The virus gains entry into cells by binding to the Ad3 receptor, which has increased expression on tumor cells. Delta-24 signifies a 24 base pair deletion in the E1a gene, leading to preferential replication in cells with defective retinoblastoma (Rb) expression, a common occurrence in tumor cells. Twenty-one patients with gynecologic malignancies received intraperitoneal injection of the OV. The treatment was well tolerated with only grade 1–2 adverse events (AE) of fatigue, malaise, and abdominal pain. The study reported that 67% achieved SD, with the remainder progressing after 1 month [72].

Ad5/3 delta 24 has been further manipulated with the addition GM-CSF gene to stimulate immune activity, with strains such as CGTG-102 and CGTG-602. In a study of CGTG-102, 60 patients with advanced solid tumors received either single of multiple dose intratumoral injections of the OV. Stable disease or better was reached in 51% of patients who received serial injections (vs. 41% in single injection). AE were minimal and limited to grade 1–2 flu-like symptoms.

## 13. Seneca Valley Virus

Seneca Valley Virus (SVV) is a more recent addition into OV group therapy. Its application focuses on tumors with neuroendocrine features, and has also proven safe in treatment of pediatric tumors [102]. In a study of 22 pediatric patients with neuroblastoma, rhabdomyosarcoma, and rare tumors with NET features, NTX-010, a SVV strain, was studied alone and in combination with cyclophosphamide. Adverse events included leukopenia, neutropenia, and tumor pain, but only 1 grade 3 toxicity (tumor pain) was reported. In another trial of SVV, SVV-001, 30 patients with advanced solid tumors with neuroendocrine features received IV injections of the OV. One patient with SCLC was progression free for 10 months [103]. This is a promising finding that warrants future study with neuroendocrine malignancies. However, a phase 2 study evaluating the efficacy of SVV in extensive stage small cell carcinoma was unfortunately negative. Non-progressive patients post 4 cycles of platinum doublet were randomized 1:1 to receive single IV infusion of SVV vs placebo. This trial was terminated due to futility as no improvement in median PFS was seen at interim analysis [127].

Recently a novel biomarker called tumor endothelial marker 8 (TEM 8) has been confirmed to act as a receptor for SVV, facilitating in viral entry in tumor cells. TEM 8 pre-screening can potentially form an important strategy to select patients for SVV therapeutic clinical trials [128].

## 14. Polio Virus

*PVSRIPO* is an oncolytic polio virus that has demonstrated promising results in the treatment of glioblastoma. It is engineered to target CD155/Necl5, the onco-fetal adhesion protein found in many solid tumors. In a study of 61 patients, the virus was injected intratumorally (Desjardins 2018) [69]. AE were minimal, with grade 4 intracranial hemorrhage occurring at only the highest injection dose. Survival is higher as compared to historical controls. Studies are underway to determine the safety of the therapy in pediatric glioma (NCT03043391) as well as invasive breast cancer (NCT03564782). It is also being studied in combination with immunotherapy for melanoma, including nivolumab (NCT04125719) and Atezolizumab (NCT03973879)

## 15. MV-NIS

MV-NIS is a unique oncolytic virus, composed of an Edmonston strain of measles virus which has been altered to express a sodium iodide transporter. Measles virus can target tumor cells through the CD46 antigen. Additionally, infection of tumor cells by the virus can be monitored through administration of Iodine-123 via SPECT imaging.

MV-NIS has been studied in the treatment of myeloma, and has exhibited promising results. In a study of 29 patients 1 CR was reported (Dispenzieri 2017) [100]. There were also some cases of decreased circulating free light chains, but they did increase once the patient cleared the virus.

## 16. Phase II Trials

Those viruses that were safe and had early clinical evidence of success progressed to Phase II studies. Phase II investigations seek to demonstrate clinical efficacy. Thirteen viruses across 24 trials were available for review. The review of Phase II data also included 6 Phase I/II trials. Again, because of limitation of space, we will discuss a few OVs used across a variety of cancers. A complete table of Phase I/II and Phase II trials in our review can be found in Table 3 and Table 4, respectively.

## 17. Reolysin

Reolysin (Pelareorep) is a strain of the reovirus type 3D that has been investigated in a variety of cancers, including ovarian, breast, melanoma, pancreatic, glioma, and multiple myeloma. Ten Phase I trials are in summarized in Table 2. This OV has provided particularly promising evidence, and use in Phase II trials have increased OS in multiple cancers. In one study of 74 patients with metastatic breast cancer, OS increased as compared to paclitaxel monotherapy (17·4 months vs. 10·4 months) [114]. In a smaller study of 14 patients with melanoma, the addition of Reolysin to paclitaxel and carboplatin saw a 21% ORR (3 patients) [111]. Reolysin in combination with gemcitabine was superior to gemcitabine monotherapy in a study of 34 patients with pancreatic cancer [113]; OS and 1- and 2-year survival was increased. Outcomes were similar to those undergoing FOLFIRINOX therapy, but the side effect profile was better tolerated with Reolysin+gemcitabine. The same success has yet to be demonstrated ovarian cancer. In a trial with 108 patients with ovarian, tubal or peritoneal cancer receiving paclitaxel, adding Reolysin did not improve outcomes [109].

## 18. ProstAtak

ProstAtak/aglatimagene besadenovec/Adv-tk is an adenovirus which has been modified to express HSV thymidine kinase. After infection of tumor cells with the virus, valcyclovir or ganciclovir is administered that subsequently induces cell death. As the name may suggest, this OV was engineered for application in prostate cancer, but has also been tested in other cancers including a Phase II trial with gliomas. In one study of 48 patients with malignant gliomas, AdV-tk was injected into the tumor cavity following resection, and subsequently received valcyclovir. Overall survival was extended in the treatment group, particularly in those patients that achieved total gross resection at time of surgery [118]. In studies that involve prostate cancer, one example included a study of 10 patients who received intraprostatic injection of Adv-tk as neoadjuvant therapy to surgery. PSA levels were monitored as a marker of tumor suppression. Eight of nine patients continued to have reduced PSA levels 10 years following injection.

## 19. ONYX-015

ONYX-015 (previously known as Ad2/5 dl1520) is an oncolytic adenovirus which was engineered to target p53-deficient cells. This adenovirus is deficient in a portion of the E1B gene that produces a 55 kDa protein. This protein typically binds to p53 to prevent apoptosis; in p53-deficient cells, the virus can replicate and lyse cancer cells. Phase II trials involved those for head and neck cancer as well as liver metastases of gastrointestinal cancers. Intratumoral injection of head and neck cancers demonstrated marked (> 50%) tumor regression in 14% of patients [135] It has also been studied in combination with 5-FU/cisplatin, for which there was 27% CR and 36% PR [119]. Hepatic artery infusion of the virus was used in 2 studies, either in combination with 5-FU/Leucovorin, or as a single agent in patients who had previously failed 5-FU/Leucovorin therapy. While there was evidence of biochemical response, increased interleukins and other immune stimulatory molecules, no clinical response was seen [37]. In the ONYX-15 single agent study [121] many patients were removed from the study due to concern about cancer progression seen on CT imaging. It was suggested alternate imaging, such as PET should be utilized in the future as this technology suggested that the increase in tumor size may have been inflammatory response from viral induced infection and inflammation.

## 20. Cavatak

Cavatak, otherwise known as CVA21, is a coxsackievirus which targets ICAM-1 to gain entry into tumor cells. Currently there is data available for studies investigating the virus’ use in advanced melanoma and non-muscle invasive bladder cancer. In a phase I study of 15 patients with non-muscle-invasive bladder cancer, patients received intravesicular injection of the virus. No serious AE occurred, and 1 CR was reported. A phase II study examined 57 patients with intratumoral injection CVA21 with advanced melanoma. There was a 21% durable response rate (CR+PR).

There are a number of current trials investigating Cavatak, predominantly in combination with immunotherapy. Its safety for use in NSCLC, melanoma, bladder cancer, and prostate cancer is being studied in the STORM trial (NCT02043665), with an additional arm in the bladder cancer and NSCLC groups to be used in combination with pembrolizumab. The CAPRA study (NCT02565992) is also combining Cavatak with pembrolizumab in metastatic melanoma. The MITCI trial (NCT02307149) is combining intratumoral Cavatak with Ipilimumab in metastatic melanoma.

## 21. Phase 3 Trials

In our systematic review, available Phase III data were limited to 2 trials of talimogene herparepvec (Table 5).

## 22. Talimogene Herparepvec

Talimogene herparepvec (T-vec) is currently the only FDA approved OV [2] available in the US. T-vec is an HSV-1 virus, which expresses human GM-CSFT-vec, that can only replicate in cancer cells. This happens because of a dysfunctional PKR pathway in cancer cells. In normal cells, viral infection causes PKR activation, which in turn, leads to an inhibition of protein synthesis within the cell and ultimately decreased viral replication. In contrast, with a dysfunction RKP system in cancer cells, viral replication proceeds unhindered [138]. However, in cancerous cells allow replication by the virus.

T-vec is indicated for unresected advance stage melanoma, administered via intratumoral injection [2]. Phase 3 trials demonstrated mixed results [36,137]. The safety profile was consistent with previous trials, with flu-like symptoms and injection site pain reported; 7% of patients had grade 3 or higher AE including nausea, vomiting, fever and abdominal pain. In the OPTiM Phase III trial [36] tumors that were not injected also demonstrated regression, which indicated possible systemic efficacy. Despite this, OS was not increased. Other studies compared T-vec to simple GM-CSF injection; T-vec produced a higher DRR (durable response rate) and increased OS than did GM-CSF alone [136].

T-vec has also been studied in combination with immunotherapy in initial phase I studies [28,139]. T-vec was injected into lesions with systemic ipilimumab therapy. Combining these two agents is more effective that either treatment alone. In another investigation, combination with pembrolizumab demonstrated increased ORR as compared pembrolizumab alone [28].

There is a Phase III trial underway examining combination therapy of systemic pembrolizumab with intratumoral injection of T-vec (NCT02263508).

## 23. Limitations of OVs

Despite the exciting evidence presented in the previously mentioned trials, OV therapy has limitations. One challenge is delivering the virus to the targeted lesions. Intratumoral injection is limited to accessible tumors such as melanomas, or targeted hepatic artery injection for hepatic lesions. While IV administration would be ideal, it can lead to sequestration of virus in the liver, limiting whole body distribution [140] as well as development of neutralizing antibodies.

Neutralizing antibodies are one of the greatest barriers to OV efficacy. Viruses selected for oncolytic virotherapy are those that can infect human cells, which carries both benefits and drawbacks for treatment. Many humans have been previously exposed to or vaccinated against some of the naturally occurring viruses used in OV therapy, and subsequently possess neutralizing antibodies to the virus. Nearly 90% of humans have antibodies against reovirus [141]. The measles virus is in development as an OV, and its success is hindered by the patient’s circulating antibodies [142].

## 24. Conclusions

The success of immune checkpoint inhibitors has propelled the field of oncology towards exploring other immunotherapeutic mechanisms to abrogate cancer growth. Oncolytic viruses are one such promising avenue with the potential to advance oncology therapeutics. While there has been only one FDA approved anticancer agent to date, a multitude of the oncolytic virus in various stages of clinical development makes for a promising story yet to unfold. The biggest advantage we have in harnessing oncolytic viral therapy lies in our capability to bioengineer these nanomicrobes. With improved understanding of molecular pathophysiology, we have the potential to manipulate oncolytic viruses to match evolving cancer challenges. 

## Figures and Tables

**Figure 1 ijms-21-07505-f001:**
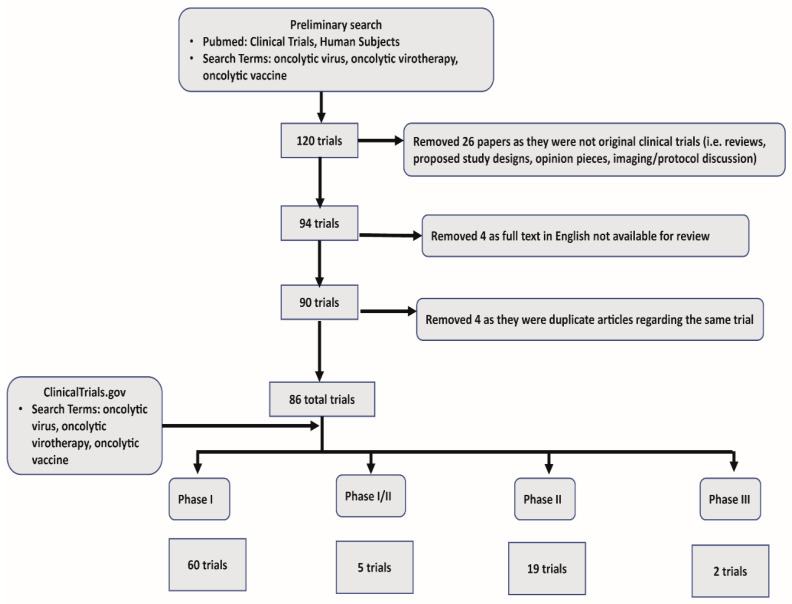
Systematic review schematics.

**Figure 2 ijms-21-07505-f002:**
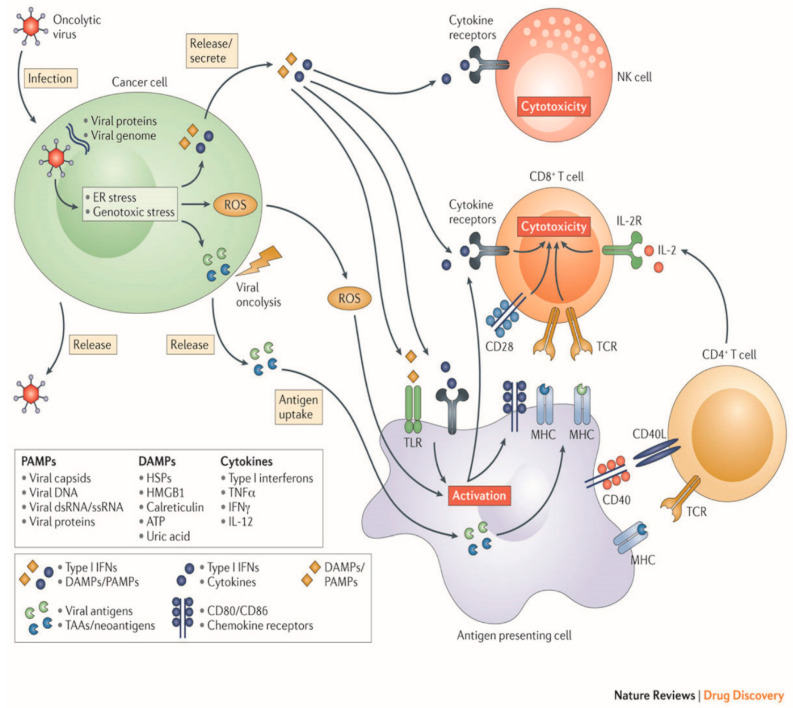
Mechanisms employed by oncolytic viruses. (Original image used with permission from Springer Nature: Kaufman, H., Kohlhapp, F. & Zloza, A. Oncolytic viruses: A new class of immunotherapy drugs. Nat Rev Drug Discov 14: 642–662 2015).

**Table 1 ijms-21-07505-t001:** Summary of combination oncolytic virus and immune checkpoint inhibitor (ici) clinical trials.

Trial Name	Virus/ICI	Cancer	*n*	Central Questions and/or Outcomes
Oncolytic Virotherapy Promotes Intratumoral T Cell Infiltration and Improves Anti-PD-1 Immunotherapy [28] (Ribas 2017)	Talimogene Laherparepvec (T-VEC) ICI: Pembrolizumab	Melanoma	21	Safety and efficacy evaluation of combination T-VEC+Pembrolizumab in melanoma. No dose limiting toxicities noted. 62% confirmed objective responses.
Randomized, Open-Label Phase II Study Evaluating the Efficacy and Safety of Talimogene Laherparepvec in Combination with Ipilimumab Versus Ipilimumab Alone in Patients with Advanced, Unresectable Melanoma [29] (Chesney 2018)	Talimogene Laherparepvec (T-VEC) ICI: Ipilimumab	Melanoma	198	Efficacy evaluation of combination Ipilimumab+T-VEC vs. Ipilimumab alone in advance melanoma. 39% Objective response rate (ORR) in combination T-VEC/Ipilimumab arm vs. 18% ORR in single agent Ipilimumab arm. 52% pts showed reduction in visceral lesions in combination arm as compared to only 23% pts in single agent ipilimumab arm.
Objective Response Rate Among Patients with Locally Advanced or Metastatic Sarcoma Treated with Talimogene Laherparepvec in Combination with Pembrolizumab [34] (Kelly 2020)	Talimogene Laherparepvec (T-VEC) ICI: Pembrolizumab	Sarcoma	20	Phase II study evaluating efficaciy of combination T-VEC and pembrolizumab in metastatic sarcoma. 35% best objective response rate. 20% grade 3 AE and no grade 4 AE noted.
LBA40-Phase Ib KEYNOTE-200: A study of an intravenously delivered oncolytic virus, coxsackievirus A21 in combination with pembrolizumab in advanced NSCLC and bladder cancer patients [32] (Rudin 2018)	Cavatak (Coxsackievirus A21) ICI: Pembrolizumab	NSCLC; Bladder Cancer	78	No DLT noted. 12% pts reported to have grade 3 treatment related adverse events. No grade 4 or 5 toxicities reported.

AE, adverse response; DLT, dose limiting toxicity; ORR, objective response rate; NSCLC non-small cell lung cancer; PD-1, programmed cell death protein 1.

**Table 2 ijms-21-07505-t002:** Summary of oncolytic virus use in phase I clinical trials.

Trial Name	Virus	Cancer	*n*	Administration, Adverse Events, Study Conclusions
Potential for efficacy of the oncolytic Herpes simplex virus 1716 in patients with oral squamous cell carcinoma [50] (Mace 2008).	*HSV1716*	Oral squamous cell carcinoma	20	Study used intratumoral injection of HSV 1716; no reportable AE, but no detectable viral replication or effective oncolysis. Study of higher doses required.
Intratumoral injection of HSV1716, an oncolytic herpes virus, is safe and shows evidence of immune response and viral replication in young cancer patients [51] (Streby 2017).	*HSV1716*	Pediatric extracranial cancers	9	Study used intratumoral injection of HSV1716; no major AE: mild constitutional symptoms (fever, chills, cyotopenia) reported. No clinical responses seen by RECIST criteria but viral replication was detected, and signs of inflammatory response was seen on PET/CT.
A phase I study of OncoVEXGM-CSF, a second-generation oncolytic herpes simplex virus expressing granulocyte macrophage colony-stimulating factor [52] (Hu 2006).	*OncoVEX^GM–CSF^ (Herpes Virus JS1/34·5-/47-/GM-CSF)*	Breast, head and neck, GI, melanoma	30	Study used intratumoral injection at different dose levels in 30 patients. AEs: fever, and injection site reaction. Three patients with SD. Some evidence of tumor necrosis was seen on biopsy which was strongly positive for HSV.
A phase 1 trial of oncolytic HSV-1, G207, given in combination with radiation for recurrent GBM demonstrates safety and radiographic responses [53] (Markert 2014).	*G207 (HSV-1)*	GBM	9	Intratumoral injection of G207 given 24 h prior to radiation. 67% with SD or PR. 3 patients with measurable response to radiation. No HSV encephalitis occurred.
A herpes oncolytic virus can be delivered via the vasculature to produce biologic changes in human colorectal cancer [54] (Fong 2009).	*NV1020*	Metastatic colon cancer in liver previously refractory to chemotherapy (5 FU and leucovorin)	12	Study used IV injection of NV1020 into the hepatic artery. CEA levels dropped in patients, and one patient saw 75% reduction of tumor volume. Study reports 7 with SD, 3 with PD, 2 patients with reduction in tumor size at 28 days. Subsequent chemotherapy was hepatic artery injection of floxuridine with dexamethasone. All patients had PR. Results confounded by varying systemic chemotherapy regimens (7 had irinotecan, 2 with oxaliplatin and 3 got both).
Intradermal injection of Newcastle disease virus-modified autologous melanoma cell lysate and interleukin-2 for adjuvant treatment of melanoma patients with resectable stage III disease [55] (Voit 2003).	Newcastle disease virus (modified with autologous melanoma cell lysate and IL-2)	Melanoma	29	Double blind study with placebo or viral injection after resection of melanoma. No clinical efficacy demonstrated.
Phase I trial of intravenous administration of PV701, an oncolytic virus, in patients with advanced solid cancers [56] (Pecora 2002).	PV701 (Newcastle virus)	Advanced solid cancers	79	Patients with advanced solid cancers refractory to traditional therapies were divided into 4 groups of differing dosing schedules (single and multiple dosing schedules). Virus was administered via IV. 62 patients were available for assessment with 1 PR and 1 CR. AE included fever, chills, nausea, hypotension.
A phase 1 clinical study of intravenous administration of PV701, an oncolytic virus, using two-step desensitization [57] (Laurie 2006).	PV701 (Newcastle virus)	Advanced solid tumors	16	Previous studies (citation) demonstrated that patients could tolerate a much higher dose of the virus if they were desensitized first, so this study executed a two-step desensitization. Minimal constitutional symptoms reported, that decreased with subsequent doses. Symptoms were also less severe than in previous studies. Study reports, 1 tumor regression, and 4 SD.
An optimized clinical regimen for the oncolytic virus PV701 [58] (Hotte 2007).	PV701 (Newcastle virus)	Advanced cancers	18	Study used IV infusions of PV701 at a slow rate. Decreasing the infusion rate allowed patients to receive higher doses of the virus, with fewer AE (particularly decreased constitutional symptoms) and minimized infusion site reaction. Study reports 1 CR, 3 PR, 2 with minor response, and 5 with SD.
Phase I trial of cyclophosphamide as an immune modulator for optimizing oncolytic reovirus delivery to solid tumors [59] (Roulstone 2015).	*RT3D (Reovirus Type 3 Dearing)*	Advanced solid tumors	36	Study used combination cyclophosphamide and RT3D (intravenously) to decrease neutralizing antibodies to the virus. While it was well tolerated, it did not reduce NARA (neutralizing antireovirus antibody) titer.
A phase I study of the combination of intravenous reovirus type 3 Dearing and gemcitabine in patients with advanced cancer [60] (Lolkema 2011).	*RT3D (Reovirus Type 3 Dearing)*	advanced solid cancers	16	First study to combine IV Reovirus with chemotherapy. AE were similar to previous studies with fever, nausea/vomiting, and chills. Protocol revised because there were grade 3 rises in LFTs (but these patients were also taking acetaminophen.) Reovirus may exacerbate gemcitabine-related liver toxicity. Clinical response was best in a patient with nasopharyngeal cancer, but OR minimal.
A phase I study of intravenous oncolytic reovirus type 3 Dearing in patients with advanced cancer [61] (Vidal 2008).	*RT3D (Reovirus Type 3 Dearing)*	Advanced cancers	36	Patients received escalating doses of IV Reovirus. There were no dose limiting toxicities, but some grade 1–2 flu-like symptoms were reported and were dose dependent. Some antitumor activity was observed by monitoring serum tumor marker levels, but not by RECIST criteria. Neutralizing Ab were detected in all patients
A phase I trial of intratumoral administration of reovirus in patients with histologically confirmed recurrent malignant gliomas [62] (Forsyth 2008).	Reolysin	Recurrent malignant gliomas	12	Study used intratumoral injection of reovirus with escalating doses. No grade 3–4 adverse events. Study reported 1 SD, 10 with PD. (1 was unable to be evaluated)
Two-stage phase I dose-escalation study of intratumoral reovirus type 3 dearing and palliative radiotherapy in patients with advanced cancers [63] (Harrington 2010)	Reolysin	Advanced solid cancers open to palliative radiation	23	Protocol used escalating doses of radiotherapy followed by intratumoral injections of RT3D. AE included flu-like symptoms (grade 2 or less) as well as neutropenia and lymphopenia. Low dose group had 2 with PR and 5 with SD, and in the high dose group 5/7 had PR and 2/7 with SD. No viral shedding seen in bodily fluids.
REO-10: a phase I study of intravenous reovirus and docetaxel in patients with advanced cancer [64] (Comins 2010).	Reolysin	Advanced solid cancers	16	Study used IV Reolysin in combination with docetaxel chemotherapy. There was 1 CR, 3 PR, and 10 with SD. Overall 88% of patients had some degree of disease control. One grade 4 neutropenia was reported.
REO-001: A phase I trial of percutaneous intralesional administration of reovirus type 3 dearing (Reolysin^®^) in patients with advanced solid tumors [65] (Morris 2013).	Reolysin	Advanced solid tumors	19	Study used intra-lesional injection of virus and was well tolerated. Most reactions were grade 1–2, including malaise, and some erythema around injection site. Study reported 37% with local tumor response, including 1 CR, 3 PR, and 4 with SD.
A phase I trial of single-agent reolysin in patients with relapsed multiple myeloma [66] (Sborov 2014)	Reolysin	Multiple myeloma	12	Virus administered intravenously. No dose limiting AE reported but grade 3 AE included hypophosphatemia, thrombocytopenia, and neutropenia. While the virus did replicate in MM cells, there was little viral protein recovered from cells. Researchers concluded it could work as part of a combination therapy, but not as monotherapy.
Phase 1 clinical trial of intratumoral reovirus infusion for the treatment of recurrent malignant gliomas in adults [67] (Kicielinski 2014)	Reolysin	Malignant glioma	15	Study used intratumoral injection. This was a dose finding study (previous study established safety). No MTD or dose limiting toxicity identified. Adverse effects related to underlying cancer, such as seizure, convulsions rather than virus. Study reported 1 PR, and a few with SD
A phase I trial and viral clearance study of reovirus (Reolysin) in children with relapsed or refractory extra-cranial solid tumors: a Children’s Oncology Group Phase I Consortium report [68] (Kolb 2015).	Reolysin	Extracranial solid tumors	24	Study used IV injection of Reolysin alone or with cyclophosphamide. No objective response to therapy, and only 1/3 of patients had a detectable viral load after 5 days; none did after 17 days. Study reports 1 grade 5 respiratory failure and 1 grade 5 thromboembolic event.
Recurrent glioblastoma treated with recombinant poliovirus [69] (Desjardins 2018)	*PVSRIPO* (Polio Virus)	Glioblastoma	61	Virus was injected intratumorally. Grade 4 ICH at the highest injection dose was the only dose-limiting toxicity. Survival rate was higher in those who received therapy as compared to historical controls, at both 24 and 36 months.
Immunological effects of low-dose cyclophosphamide in cancer patients treated with oncolytic adenovirus [70] (Cerullo 2011).	*Ad5/3-(delta)24*	Advanced solid tumors resistant to chemotherapy	21	Study used intratumoral injection with adenovirus followed by cyclophosphamide treatment in different dosing groups. AE were mostly grade 1–2 constitutional symptoms. The one year PFS and OS was increased compared to traditional chemotherapy resistant cancers. Study reports 8/12 patients with RECIST response: 2 with MR, 6 with SD, and 4 with PD.
A phase I clinical trial of Ad5/3-Δ24, a novel serotype-chimeric, infectivity-enhanced, conditionally-replicative adenovirus (CRAd), in patients with recurrent ovarian cancer [71] (Kim 2013).	*Ad5/3-(delta)24*	Ovarian Cancer	9	Study used intraperitoneal injection of the virus. AE were flu-like grade 1–2: fever/chills, myalgias, fatigue, and nausea. Study reported 3 patients with a decrease in CA-125 levels at 1 month.
A phase I study of a tropism-modified conditionally replicative adenovirus for recurrent malignant gynecologic diseases [72] (Kimball 2010).	*Ad5/3-(delta)24*	Gynecologic malignancy	21	Study used intraperitoneal injection of virus. Reported grade 1-2 adverse events included fatigue, fever an abdominal pain. Study reported 71% of patients had SD, and the remainder had PD at 1 month. No CR or PR were achieved.
Integrin targeted oncolytic adenoviruses Ad5-D24-RGD and Ad5-RGD-D24-GMCSF for treatment of patients with advanced chemotherapy refractory solid tumors [73] (Pesonen 2012).	*Ad5-D24-RGD and Ad5-RGD-D24-GMCSF*	advanced solid tumors resistant to chemotherapy	16	16 patients were injected with adenovirus, 9 treated with Ad5-D24-RGD, and 7 treated with Ad5-RGD-D24-GMCSF. Virus for one group contained GM-CSF; as large tumors have immunosuppressive characteristics, GM-CSF might stimulate the immune system. Some patients in the GMCSF group showed SD, while all patients in the other group progressed. AE were low grade (1–2); constitutional symptoms or injection site pain.
Antiviral and antitumor T cell immunity in patients treated with GM-CSF-coding oncolytic adenovirus [74] (Kanerva 2013).	*CGTG-102 (Ad5/3-delta24-GMCSF)*	Advanced solid tumors	60	60 patients received intratumoral injections. The study compared single injection (39 patients) to multiple injections (21 patients) to establish safety of multiple. Stable disease or better was achieved in 50% with serial injection vs. 41% with single injection. Mostly grade 1–2 AE occurred (constitutional symptoms).
Immunological data from cancer patients treated with Ad5/3-E2F-Δ24-GMCSF suggests utility for tumor immunotherapy [75] (Hemminki 2015).	*CGTG-602 (Ad5/3-E2F-delta24-GMCSF)*	Advanced solid tumors	13	13 patients enrolled with varying doses and treatment rounds of virus via IV. 50% of patients noted a response to treatment. AE were predominately grade 1–2 constitutional symptoms, but some grade 3 AE were seen.
A phase I trial of CV706, a replication-competent, PSA selective oncolytic adenovirus, for the treatment of locally recurrent prostate cancer following radiation therapy [76] (DeWeese 2001).	CV706 (PSA selective adenovirus)	Prostate cancer	20	Study used intraprostatic injection of virus into patients with locally recurrent prostate cancer. There were no grade 3 or greater toxicities. There was evidence of replication in biopsy tissues. Those treated with higher doses of the virus had at least 50% drop in PSA levels.
A phase I trial of intravenous CG7870, a replication-selective, prostate-specific antigen-targeted oncolytic adenovirus, for the treatment of hormone-refractory, metastatic prostate cancer [77] (Small 2006).	*CG7870*	Hormone refractory metastatic prostate cancer	23	Patients received intravenous dosing of the virus. Most AE were constitutional symptoms such as fatigue, fevers, nausea. Three grade 3 reactions occurred, including severe fatigue. MTD was reached due to transaminitis and elevated d-dimer levels. Using PSA as the endpoint, study reported 5 patients with PSA reduction of 25–49% after 1 treatment, but no PR or CR were reported.
A first in human phase 1 study of CG0070, a GM-CSF expressing oncolytic adenovirus, for the treatment of nonmuscle invasive bladder cancer [78] (Burke 2012).	*CG0070 (GM-CSF expressing adenovirus)*	Bladder cancer (non-muscle)	35	Patients received intravesical infusions of virus. Grade 1–2 bladder toxicities were the most frequent AE but 3 patients had grade 3 reactions for nocturia, dysuria and lymphopenia. Study reported CR of 48.6% across all groups, and higher (58.3%) in those with high Rb phosphorylation.
A Phase I study of KH901, a conditionally replicating granulocyte-macrophage colony-stimulating factor: armed oncolytic adenovirus for the treatment of head and neck cancers [79] (Chang 2009).	*KH901 (GM-GSF Expressing Adenovirus)*	recurrent head and neck cancers	23	KH901, selective for cells expressing telomerase, was injected intratumorally in patients with recurrent head and neck cancer. Groups included single, and multi-dose injections at escalating dosages. There were was no dose-limiting AE; majority were grade 1-2 constitutional symptoms. Study reports 7/19 patients with PD, 12 with SD.
Oncolytic adenovirus ICOVIR-7 in patients with advanced and refractory solid tumors [80] (Nokisalmi 2010).	*ICOVIR-7 (adenovirus)*	Advanced solid tumors	21	Study used intratumoral injection at varying doses. One grade 3 anemia was observed, while remaining side effects were grade 1–2 and included flu-like symptoms, increased liver transaminases, and hyponatremia. Study reported 5 objective responses including 1 PR, 2 minor, and 2 SD. All patients had PD prior to trial initiation.
A phase I trial of intratumoral administration of recombinant oncolytic adenovirus overexpressing HSP70 in advanced solid tumor patients [81] (Li 2009).	Telomelysin/OBP-301 (adenovirus with human telomerase reverse transcriptase, hTERT)	Advanced solid tumors	27	Study used intratumoral injection of virus in a dose escalation phase I study. Rare grade III fever and grave IV thrombocytopenia at high doses were observed. Most AE were fever, reaction at the injection site, as well as thrombocytopenia, and depressed leukocyte and lymphocyte counts. Study reported a minimum of 48% with SD, and 11% had CR or PR.
A phase I open-label, dose-escalation, multi-institutional trial of injection with an E1B-Attenuated adenovirus, ONYX-015, into the peritumoral region of recurrent malignant gliomas, in the adjuvant setting [82] (Chiocca 2004).	*ONYX-015*	Malignant glioma	24	24 patients received varying doses of ONYX-015 injected into 10 areas of the resected glioma cavity. No severe AE were reported that were likely related to treatment (10 patient did have AE, including 1 grade 3–4 neuropathy) Only 1 patient did not have PD and No real treatment effect could be correlated with the viral treatment.
A phase I trial of intravenous infusion of ONYX-015 and Enbrel in solid tumor patients [83] (Nemunaitis 2007).	*ONYX-015*	Advanced cancers	9	Nine patients divided in 3 groups received IV infusion of onyx-015 of varying doses, with a dose of Enbrel. Study reported 4/9 with SD, but no regression was seen. AE were mild. Circulating viral DNA was higher when virus infusion is given in combination with Enbrel
A phase I study of Onyx-015, an E1B attenuated adenovirus, administered intratumorally to patients with recurrent head and neck cancer [84] (Ganly 2000).	*ONYX-15*	Recurrent head and neck cancer	22	Study used single intratumoral injection that was well tolerated. Grade 1–2 constitutional symptoms were most common. Study reported no OR by RECIST criteria, but evidence of tumor necrosis seen on MRI in 5 patients with questionable PR was noted.
A phase I study of telomerase-specific replication competent oncolytic adenovirus (telomelysin) for various solid tumors [85] (Nemunaitis 2010).	*H103 (Adenovirus expressing HSP70)*	Advanced solid tumors	27	Study used intratumoral injection of virus in a dose escalation phase I study. Rare grade III fever and grave IV thrombocytopenia at high doses was observed. Most AE were fever, and a local reaction at the injection site, as well as thrombocytopenia, and depressed leukocyte and lymphocyte counts. Study reported 48% of patients had at least SD or better, and 11% had CR or PR.
Phase 1 study of intravenous administration of the chimeric adenovirus enadenotucirev in patients undergoing primary tumor resection [86] (Garcia-Carbonero 2017).	*Enadenotucirev (adenovirus aka ColoAd1)*	Colorectal, NSCLC, Urothelial, RCC	17	Study used IV infusion for NSCLC, RCC and urothelial cancers, and intrathehecal injection in colorectal cancer. Both demonstrated high local CD8+ cell infiltration, with no significant treatment-related AE.
Phase I study of replication-competent adenovirus-mediated double suicide gene therapy for the treatment of locally recurrent prostate cancer [87] (Freytag 2002). Five-year follow-up of trial of replication-competent adenovirus-mediated suicide gene therapy for treatment of prostate cancer [88] (Freytag 2007).	*Ad5-CD/TKrep*	Prostate (recurrent)	16	Patients were injected with virus, and two days later received ganciclovir and 5-fluorocytosine prodrug. Study reported >25% decreased PSA levels in 44% of patients. A 5-year follow up showed PSA doubling time was extended in patients who received the virus treatment indicating that patients had longer until salvage treatment was needed.
Phase I study of replication-competent adenovirus-mediated double-suicide gene therapy in combination with conventional-dose three-dimensional conformal radiation therapy for the treatment of newly diagnosed, intermediate- to high-risk prostate cancer [89] (Freytag 2003).	*Ad5-CD/TKrep*	Prostate	15	Patients had intraprostatic injection of adenovirus with cytosine deaminase and HSV thymidine kinase genes. 2 days later the patients received 5-fluorocytosine and valganciclovir prodrug for up to 4 weeks along with radiation. PSA ½ life was decreased in those with more than 1 week of prodrug therapy. 94% of AE were mild to moderate; severe reactions were similar to reactions obtained with standard radiation therapy.
Phase I trial of replication-competent adenovirus-mediated suicide gene therapy combined with IMRT for prostate cancer [90] (Freytag 2007).	*Ad5-yCD/mutTKSR39rep-ADP*	Prostate	9	This study used a second generation virus with improved enzyme activity administered via intraprostatic injection. Reported AE include 13% with grade 3 lymphopenia; other AE were grade 1–2. Prostate biopsies at the end of the trial had fewer positives for residual adenocarcinoma than expected (22% rather than >40%)
Use of a targeted oncolytic poxvirus, JX-594, in patients with refractory primary or metastatic liver cancer: a phase I trial [91] (Park 2008).	*Pexa-Vec/JX-594 (pexastimogene devacirepvec)*	Primary or metastatic liver cancer	14	This was a dose-escalation study of intratumoral injection of JX-594. Grade 3 hyperbilirubinemia occurred in patients with the highest dose. All experienced flu-like symptoms, ranging grade I–III and 4 had short-lived grade I–III dose dependent thrombocytopenia. Study reported response in the injected as well as non-injected tumors; 3 PR, 6 with SD and 1 with PD (and 4 could not be evaluated via imaging for different reasons).
A mechanistic proof-of-concept clinical trial with JX-594, a targeted multi-mechanistic oncolytic poxvirus, in patients with metastatic melanoma [92] (Hwang 2011).	*Pexa-Vec/JX-594 (pexastimogene devacirepvec)*	Metastatic melanoma	10	Patients were injected intratumorally with 1/10^th^ the dose of normal JX-594. Biopsies demonstrated evidence of tumor necrosis, as well as gene expression from JX-594. Clinical outcomes were not reported (not focus of study). Mild constitutional symptoms as previously reported were the only AE.
Phase 1 study of intratumoral Pexa-Vec (JX-594), an oncolytic and immunotherapeutic vaccinia virus, in pediatric cancer patients [93] (Cripe 2015).	*Pexa-Vec/JX-594 (pexastimogene devacirepvec)*	neuroblastoma, Ewing sarcoma, HCC	3	Intratumoral injection of Pexa-Vec in 3 patients. All 3 developed skin pustules (grade 1) that lasted 3–4 weeks. Study reported no OR by RECIST criteria; one patient had evidence of tumor necrosis on imaging.
Phase 1b trial of biweekly intravenous Pexa-Vec (JX-594), an oncolytic and immunotherapeutic vaccinia virus in colorectal cancer [94] (Park 2015).	*Pexa-Vec/JX-594 (pexastimogene devacirepvec)*	Colorectal	15	Study used IV infusion of Pexa-Vec in 15 patients. AE were grade 1–2, and mostly constitutional symptoms such as fever, malaise, chills, myalgias. Study reported 67% patients with SD as seen on imaging.
Vectorized gene therapy of liver tumors: proof-of-concept of TG4023 (MVA-FCU1) in combination with flucytosine [95] (Husseini 2017).	*TG4023 (MVA-FCU1 modified vaccinia virus)*	Primary or metastatic liver tumors resistant to other forms of treatment	16	TG4023 contains the gene for an enzyme to convert flucytosine into cytotoxic 5-fluorouracil. 16 patients had intratumoral injection at escalating doses, and then the prodrug (flucytosine) on day 2. Tumor biopsy demonstrated therapeutic levels of the active drug were reached. 7 patients had dose limiting toxicity with non-sustained rise in AST and ALT. Most frequent AE were constitutional symptoms including anorexia, fever, and fatigue. Study reported 50% of patients with SD, the other half with PD.
Phase I trial of intravenous oncolytic vaccinia virus (GL-ONC1) with cisplatin and radiotherapy in patients with locoregionally advanced head and neck carcinoma [96] (Mell 2017).	*GL-ONC1 (Vaccinia Virus)*	Head and Neck Cancer (locoregionally advances without metastasis)	19	Study used IV injection of virus along with cisplatin and radiotherapy. Most AE were grade 1–2 constitutional symptoms and rash, but 2 patients had grade 3 hypotension, nausea/vomiting and mucositis. Study reported 5/19 patients had virus present in the biopsy. At 1 year/2 years: 74.4%/64.1% with PFS and 84.6%/69.2% OS
First-in-man study of western reserve strain oncolytic vaccinia virus: safety, systemic spread, and antitumor activity [97] (Zeh 2015).	*vvDD (Poxvirus – western reserve strain oncolytic vaccinia virus)*	Advanced solid tumors	16	Study used intratumoral injection of virus. Selective viral replication reported in both injected and non-injected rumors. Study reported 1 grade 3 events occurred, which was pain in a breast cancer patient around the time of highest inflammation.
Phase 1 Study of intravenous oncolytic Poxvirus (vvDD) in patients with advanced solid cancers [98] (Downs-Canner 2016).	*vvDD (Poxvirus – western reserve strain oncolytic vaccinia virus)*	Advanced colorectal or other solid cancers	11	Study used IV administration. No dose limiting toxicities reported, and most AE were grade 1 or 2 constitutional symptoms. Study reported Th1 cytokines and inflammatory reaction occurred. A mixed response on some liver metastasis with improvement of cutaneous melanoma.
Oncolytic measles virus in cutaneous T cell lymphomas mounts antitumor immune responses in vivo and targets interferon-resistant tumor cells [99] (Heinzerling 2005).	*MV (Measles Virus, Edmonston-Zagreb strain)*	Cutaneous T cell Lymphoma	5	Study used intratumoral injections of live virus with dose escalation. Endpoint was TBI (Tumor Burden Index: 1 lesion resolved (CR); 2 showed evidence of regression in local but non-injected lesions; remaining tumors unchanged. AE included grade 1 injection site erythema, arthralgias, itching only
Phase I trial of systemic administration of Edmonston strain of measles virus genetically engineered to express the sodium iodide symporter in patients with recurrent or refractory multiple myeloma [100] (Dispenzieri 2017).	*MV-NIS (measles virus with sodium iodide symporter)*	Relapsed and refractory multiple myeloma	29	Study used IV injection; one group with MV-NIS alone, another with cyclophosphamide prior to MV-NIS treatment. Study reported grade 3–4 hematologic AEs including decreased blood counts neutropenia, thrombocytopenia, anemia, and lymphopenia for both groups. 1 CR reported with some cases of short lived decreased circulating free light chains (increased once the body cleared the virus). Iodine was used to identify infection in myeloma cells.
Phase I trial of intraperitoneal administration of an oncolytic measles virus strain engineered to express carcinoembryonic antigen for recurrent ovarian cancer [101] (Galanis 2010).	*MV-CEA (Measles virus, Edmonston strain)*	Taxol and platinum-refractory recurrent ovarian with normal CEA levels	21	Study used intraperitoneal injection of the virus with CEA as measure of viral replication. Study reports best response (per RECIST) was SD in 14/21 and was dose dependent; 5 patients had marked decrease of CA-125 levels. Median survival was 12·15 compared to 6 months for historical controls.
Phase I trial of Seneca Valley Virus (NTX-010) in children with relapsed/refractory solid tumors: a report of the Children’s Oncology Group [102] (Burke 2015).	*NTX-010 (Seneca Valley Virus)*	Pediatric patients wih neuroblastoma, rhabdomyosarcoma, rare tumors with NET features	22	Study designed in 2 parts: part A was dose finding, virus only, using 3 doses; part B added cyclophosphamide. Study reported AEs were leukopenia, neutropenia; tumor pain in 1 patient was the only grade 3 toxicity observed. Nearly all patients (17/18) had neutralizing antibodies which prohibited virus efficacy.
Phase I clinical study of Seneca Valley Virus (SVV-001), a replication-competent picornavirus, in advanced solid tumors with neuroendocrine features [103] (Rudin 2011).	*SVV-001 (Seneca Valley Virus, a picornavirus)*	Advanced solid tumors with neuroendocrine features	30	Study used IV injection of virus. 1 patient with SCLC was progression free at 10 months. AEs included flu like symptoms (fever, fatigue headache) and one grade 3 lymphopenia
A phase I dose-escalation clinical trial of intraoperative direct intratumoral injection of HF10 oncolytic virus in non-resectable patients with advanced pancreatic cancer [104] (Nakao 2011).	*HF-10*	Pancreatic cancer	6	Study used 3 intratumoral injections of HF-10. Study reported PD in 2 patients, SD in 3 patients, and 1 had a PR. No AE reported.
A Phase I clinical trial of EUS-guided intratumoral injection of the oncolytic virus, HF10 for unresectable locally advanced pancreatic cancer [105] (Hirooka 2018).	*HF-10*	Pancreatic cancer	10	Study used EUS injection of HF10 for unresectable pancreatic cancer up to 4 times, every 2 weeks. Co-treatment was erlotinib and gemcitabine. Study reported AEs: 5 patients with severe myelosuppression; 2 patients had severe events not due to HF10. Study outcomes: 2 PD, 4 SD, and 3 PR. PFS 6·3 months, OS 15·5 months. 2 patients achieved CR after downstaging and surgery.
Results of a randomized phase I gene therapy clinical trial of nononcolytic fowlpox viruses encoding T cell costimulatory molecules [106] (Kaufman 2014).	*rF-B7.1 and rF-TRICOM (recombinant fowlpox virus with either B7.1 or three genes: B7.1, ICAM-1, and LFA-3)*	Melanoma and colon cancer	12	Study used intratumoral injection of one of the two viruses and at varying doses, every 4 weeks. AE were minimal, including injection site pain and pyrexia (in only 4 patients). Study reported no objective clinical responses but safety was established, and some T cell activity specific to the tumors was seen Stable disease noted for 3 patients which included both colon cancer cases.

AE, adverse event; CR, complete response; GI, gastrointestinal; ICH, intracerebral hemorrhage; LFT, liver function test; MM, multiple myeloma; MR, marginal response; MTD, maximum tolerated dose; NARA, neutralizing antivirus antibody; OR, overall response; ORR, objective response rate; OS, overall survival; PD, progressive disease; PFS, progression free survival; PR, partial response; RECIST, response evaluation criteria in solid tumors; SCLC, small cell lung cancer SD, stable disease.

**Table 3 ijms-21-07505-t003:** Summary of oncolytic virus use in phase II clinical trials.

Trial Name	Virus	Cancer	*n*	Outcomes
Randomized phase IIB evaluation of weekly paclitaxel versus weekly paclitaxel with oncolytic reovirus (Reolysin^®^) in recurrent ovarian, tubal, or peritoneal cancer [109] (Cohn 2017).	Reolysin	Ovarian, tubal or peritoneal cancer	108	Study reported Reolysin did not improve outcomes enough to induce further study.
Phase II trial of intravenous administration of Reolysin(^®^) (Reovirus Serotype-3-dearing Strain) in patients with metastatic melanoma [110] (Galanis 2012).	Reolysin	Melanoma	21	Viral replication confirmed in biopsies; 1 patient had 75–90% tumor necrosis, which provided evidence for treatment effect.
A phase II study of REOLYSIN^®^ (pelareorep) in combination with carboplatin and paclitaxel for patients with advanced malignant melanoma [111] (Mahalingam 2017).	Reolysin	Melanoma	14	Study reported 3 partial responses: 1% ORR; PFS 5·2 months; OS 10·9 months. 1 year OS was 43%. Disease control rate 85%.
Randomized phase 2 trial of the oncolytic virus Pelareorep (Reolysin) in upfront treatment of metastatic pancreatic adenocarcinoma [112] (Noonan 2016).	Reolysin	Metastatic pancreatic adenocarcinoma	73	Study reported that addition of Reolysin to paclitaxel+carboplatin, did not improve PFS, but was well-tolerated. Presence of KRAS mutation also did not affect outcome.
A Phase II study of Pelareorep (REOLYSIN^®^) in combination with gemcitabine for patients with advanced pancreatic adenocarcinoma [113] (Mahalingam 2018).	Reolysin	Pancreatic adenocarcinoma	34	Study reported OS (10·2 vs. 6·8 months) as well as 1- and 2-year survival (45% and 22%, respectively) was increased compared to historical controls that used single agent gemcitabine (20–22%, 2·5%).
A randomized phase II study of weekly paclitaxel with or without pelareorep in patients with metastatic breast cancer: final analysis of Canadian Cancer Trials Group IND.213 [114] (Bernstein 2018).	Reolysin	Metastatic breast cancer	74	There was no statistical difference in RR or PFS but there was increased OS: 17·4 months vs. 10·4 months with paclitaxel alone.
Prospective randomized phase 2 trial of intensity modulated radiation therapy with or without oncolytic adenovirus-mediated cytotoxic gene therapy in intermediate-risk prostate cancer [115] (Freytag 2014).	*Ad5-yCD/* *mut* *TK_SR39_* *rep* *-ADP adenovirus*	Prostate	44	Combined intensity modulated radiation therapy (IMRT) with adenovirus. Significant decrease in the number of positive biopsies at 2 years: IMRT 58%; IMRT+virotherapy 33% (*P* = 0.13).
Intraprostatic distribution and long-term follow-up after AdV-tk immunotherapy as neoadjuvant to surgery in patients with prostate cancer [116] (Rojas-Martinez, 2013).	*AdV-tk (aka aglatimagene besadenovec or ProstAtak*	Prostate	10	Injected AdV-tk with valcyclovir or ganciclovir in 9 patients scheduled for prostatectomy. 6 of these patients had high risk features. At 10 years did not have signs of recurrence/metastasis. Historic controls with high risk features typically have 15% PSA failure with 34% of those have metastasis [117] (Pound 1999)
Phase II multicenter study of gene-mediated cytotoxic immunotherapy as adjuvant to surgical resection for newly diagnosed malignant glioma [118] (Wheeler 2016).	*AdV-tk (aka aglatimagene besadenovec or ProstAtak*	Glioma	48	Patients with malignant gliomas were injected AdV-tk during surgery. They then received valcyclovir along with standard therapy of radiation and temozolomide. Overall survival was extended for those who received viral therapy (especially in the group in which total gross resection was achieved at the initial surgery).
A controlled trial of intratumoral ONYX-015, a selectively-replicating adenovirus, in combination with cisplatin and 5-fluorouracil in patients with recurrent head and neck cancer [119] (Khuri 2000).	*ONYX-15*	Recurrent squamous cell head and neck cancer	37	Patients were given cisplatin/5 FU along with intratumoral injection of ONYX-015. Study reported 27% CR and 36% PR. There were flu-like symptoms and injection site pain most commonly, and rare grade 3–4 mucositis.
Selective replication and oncolysis in p53 mutant tumors with ONYX-015, an E1B-55kD gene-deleted adenovirus, in patients with advanced head and neck cancer: a phase II trial [120] (Nemunaitis 2000).	*ONYX-15*	Recurrent head and neck cancer	37	Intratumoral injection of virus led to marked tumor regression (as defined as >50% of lesion) in 21% of patients. Biopsy of surrounding normal tissues were negative for viral infection.
Hepatic arterial infusion of a replication-selective oncolytic adenovirus (dl1520): phase II viral, immunologic, and clinical endpoints [37] (Reid 2002).	*Onyx-015 (Adenovirus dl1520)*	Gastrointestinal cancer with liver metastasis	27	Study used hepatic artery infusion of the virus in combination with traditional chemotherapy (5-FU and leucovorin). Study reported biochemical response with virotherapy (increase TNF, IFN-gamma, IL-6 and IL-10) in some tumors resistant to chemotherapy alone. Some grade 3–4 adverse events were seen with hyperbilirubinemia.
Effects of Onyx-015 among metastatic colorectal cancer patients that have failed prior treatment with 5-FU/leucovorin [121] (Reid 2005).	*Onyx-015 (Adenovirus dl1520)*	Colorectal cancer with metastasis to liver	24	Study used hepatic artery infusion of virus in subjects that failed 5-FU/leucovorin. Study reported mixed results: some evidence of tumor necrosis and regression; many patients removed early due to CT-demonstrated enlargement of tumors. However, CT information may reflect inflammation in response to tumor infection with virus prior to regression; PET rather than CT use suggested in the future.
An open label, single-arm, phase II multicenter study of the safety and efficacy of CG0070 oncolytic vector regimen in patients with BCG-unresponsive non-muscle-invasive bladder cancer: Interim results [122] (Packiam 2017).	*CG0070 (GM-CSF expressing adenovirus)*	Non-muscle invasive bladder cancer (NMIBC),	45	Study used intravesical CG0070 for those resistant to bacillus Calmette–Guerin (BCG). Study reported a 47% CR at 6 months.
Randomized dose-finding clinical trial of oncolytic immunotherapeutic vaccinia JX-594 in liver cancer [108] (Heo 2013),	*Pexa-Vec (JX-594 vaccinia virus)*	HCC	22	mRECIST and Choi tumor response rates were equivalent for the low and high dose treatment groups. Median survival was greater in the high dose group (14·1 months vs. 6·7 months).
Phase II trial of Pexa-Vec (pexastimogene devacirepvec; JX-594), an oncolytic and immunotherapeutic vaccinia virus, followed by sorafenib in patients with advanced hepatocellular carcinoma (HCC) [123] (Heo 2013).	*Pexa-Vec (JX-594 vaccinia virus)*	HCC	25	Study used Pexa-Vec IV for Day 1 followed by intratumoral injections on Days 8 and 22; sorafenib was administered on Day 25. Per mRECIST criteria, 62% had disease control after Pexa-vec treatment, and 59% after use of sorafenib.
Phase II clinical trial of a granulocyte-macrophage colony-stimulating factor-encoding, second-generation oncolytic herpesvirus in patients with unresectable metastatic melanoma [35] (Senzer 2009).	*OncoVEX^GM–CSF^ (Herpes Virus JS1/34·5-/47-/GM-CSF)*	Melanoma	50	Study used intratumoral injection of the virus. Study reported a 26% response rate using the RECIST criteria which included uninjected tumors (and even visceral tumors).
Phase II clinical study of intratumoral H101, an E1B deleted adenovirus, in combination with chemotherapy in patients with cancer [124] (Xu 2003).	*H101 (E1B deleted adenovirus)*	Malignant tumors	50	Study used intratumoral injection 5 days/week for 3 weeks along with standard chemotherapy. Response rate was 28% vs 13% in control group. Grade 4 hematologic AE in 4 patients.
Phase I Trial of an ICAM-1-Targeted Immunotherapeutic-Coxsackievirus A21 (CVA21) as an Oncolytic Agent Against Non Muscle-Invasive Bladder Cancer [125] (Annels 2019)	Coxsackievirus A21 (CVA21)	Non-muscle invasive bladder cancer	15	Nine patients received intravesicular injection of virus; 6 patients received viral injection and subtherapeutic mitotycin C before resection. 1 CR (by histology) No serious AE reported.
A Study of Intratumoral CAVATAK™ in Patients with Stage IIIc and Stage IV Malignant Melanoma (VLA-007 CALM) (CALM) [126] (NCT01227551)	Coxsackievirus A21 (CVA21)	Stage IIIc and Stage IV Malignant Melanoma	57	Intratumoral injection of virus. Reported 38·6% immune-related Progression-Free Survival at 6 months, 21% durable response rate (CR+PR). 19% experience severe AE.

5-FU, fluorouracil; AE, adverse response; BCG, bacillus Calmette–Guerin; CT, computerized tomography; IL-6/IL10, interleukin 6/interleukin 10; IMRT, intensity modulated radiation therapy; IFN, interferon; mRECIST, modified RECST; ORR, objective response rateOS, overall survival; PET, positron emission tomography; PFS, progression free survival; RECIST, response evaluation criteria in solid tumors; RR, response rate; TNF, tumor necrosis factor.

**Table 4 ijms-21-07505-t004:** Summary of oncolytic virus use in Phase I/II studies.

Trial Name	Virus	Cancer	*n*	Administration, Adverse Events, Study Conclusions
Phase I/II trial of carboplatin and paclitaxel chemotherapy in combination with intravenous oncolytic reovirus in patients with advanced malignancies [129] (Karapanagiotou 2012).	RT3D (Reovirus Type 3 Dearing)	Advanced head and neck cancer	31	Study used IV infusion of RT3D with carboplatin/paclitaxel. Study reported treatment was well tolerated, no MTD, and toxicities were largely grade I/II. Outcomes reported by RECIST were 1 CR (3·8%) and 6 PR (23·1%).
Oncolytic H-1 parvovirus shows safety and signs of immunogenic activity in a first phase I/IIa glioblastoma trial [130] (Geletneky 2017).	Parvovirus	Glioblastoma	18	Study used IV and intratumoral injection into cavity created from resection. Study reported: no MTD was reached; T cell infiltration and activation of macrophages and microglia were detected in the infected tumors (Evidence that it is stimulating the immune system) Median survival increased compared to recent meta-analyses.
Phase I/II study of oncolytic HSV GM-CSF in combination with radiotherapy and cisplatin in untreated stage III/IV squamous cell cancer of the head and neck [131] (Harrington 2010).	HSV GM-CSF	Stage III/IV squamous cell cancer of head and neck	17	Study used intratumoral injection every 21 days with radiation and cisplatin; neck dissection 6–10 weeks later. Study reported a median follow up of 29 months with 76·5% relapse free rate, 100% locoregional control (60–70% in historical controls).
Phase I/II study of oncolytic herpes simplex virus NV1020 in patients with extensively pretreated refractory colorectal cancer metastatic to the liver [132] (Geevarghese 2010).	NV1020 (Herpes Virus)	Colorectal liver metastases	I 13 II 19	Study used hepatic arterial injections with virus then standard chemotherapy in patients with relapse. Study documents 50% SD with a median survival longer than historical controls.
Phase I–II trial of ONYX-015 in combination with MAP chemotherapy in patients with advanced sarcomas [133] (Galanis 2005).	*Onyx-015 (Adenovirus dl1520)*	Advanced sarcomas	6	Study used intratumoral injection of the virus in combination with MAP applied to metastases in the liver and chest wall. No significant toxicities identified. Study outcomes include:1 patient with PR that lasted 11 months; tumor had p53 mutation and MDM-2 amplification
Phase I/II trial of intravenous NDV-HUJ oncolytic virus in recurrent glioblastoma multiforme [134] (Freeman 2006).	*NDV-HUJ (HUJ strain of Newcastle disease virus)*	glioblastoma multiforem	14	Study used IV injection of virus. Study reported AEs grade I/II constitutional symptoms. 1 person with CR.

CR, complete response; MAP, mitomycin-C+ doxorubicin+cisplatin; MTD, maximum tolerated dose; OS, overall survival; PR, partial response; RECIST, response evaluation criteria in solid tumors; SD, stable disease.

**Table 5 ijms-21-07505-t005:** Summary of oncolytic virus use in Phase III clinical trials.

Trial Name	Virus	Cancer	*n*	Central Questions and/or Outcomes
A Study of Talimogene Laherparepvec in stage IIIc and stage IV malignant melanoma [136] (Andtbacka 2015).	Talimogene Laherparepvec	Stage IIB-IV melanoma	295	T-VEC was compared to subcutaneous GM-CSF. Higher DRR at 6 months and greater median OS with T-VEC particularly in untreated advanced melanoma. Lesions that were not directly injected showed response. OS was not improved.
Phase IIIb safety results from an expanded-access protocol of talimogene laherparepvec for patients with unresected, stage IIIB-IVM1c melanoma [137] (Chesney 2018).	Talimogene Laherparepvec	Melanoma	41	Safety profile was consistent with previous trials. As compared to OPTIM trial above, this trial also included ECOG of 2 (OPTIM was 0, 1 only). Efficacy was not assessed, as the primary outcome was to provide expand access to T-VEC until FDA approval.

AE, adverse response; DRR, durable response rate; OS, overall survival.

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
