# Peer review of "Clinical Application of Oncolytic Viruses: A Systematic Review"

_ijms, 2020, doi:10.3390/ijms21207505_

Round 1

Reviewer 1 Report

The authors have done an extensive literature search and summarized all of the available published papers on the clinical studies of oncolytic viruses. With a few misses, they have done a very good job.

As the most promising trials would be combination therapy with oncolytic viruses, especially with immune checkpoint blockade.  One suggestion would be to list a New Section “Combination therapy trials”.  There are two key clinical trials in which the oncolytic virus T-VEC was combined with either anti-PD-1 and anti-CTLA-4 ab. The combination enhanced the clinical response rates from ~20% up to about 65% in patients with advanced melanoma.

(1). Ribas A et al., Cell 2017 (ref #59); (2). Chesney J et al., J Clin Oncol. 2018; 36:1658-67.

The authors have cited one (ref#59).

Minor issues:

  1. Page 6, lines 201-204. The authors missed an important fact: This phase III trial with Pexa-vec has been terminated in August of 2019, due to ineffctivenee.

https://www.cancertherapyadvisor.com/home/cancer-topics/general-oncology/phase-3-trial-for-oncolytic-viral-therapy-pexa-vec-in-advanced-liver-cancer-terminated-early/

  1. Page 6. Line 219: there is a typo. It is “Senaca Valley virus”, not “Senava…”.
  2.  
  3. Page 7. Section of Phase II trial. The authors missed a phase II trial with NTX-010 (the Seneca Valley virus). It is Schenk EL et al. J Thorac Oncol. 2020; 15: 110-119.

Author Response

Response to Reviewers

Dear Editors:

We would like to thank the editors and reviewers for their detailed and valuable comments. We believe we have addressed all the concerns and hope the corrections will be found satisfactory. We are however happy to make further changes if needed.

Reviewer #1 

Comments and Suggestions for Authors:

The authors have done an extensive literature search and summarized all of the available published papers on the clinical studies of oncolytic viruses. With a few misses, they have done a very good job.

As the most promising trials would be combination therapy with oncolytic viruses, especially with immune checkpoint blockade.  One suggestion would be to list a New Section “Combination therapy trials”.  There are two key clinical trials in which the oncolytic virus T-VEC was combined with either anti-PD-1 and anti-CTLA-4 ab. The combination enhanced the clinical response rates from ~20% up to about 65% in patients with advanced melanoma.

(1). Ribas A et al., Cell 2017 (ref #59); (2). Chesney J et al., J Clin Oncol. 2018; 36:1658-67.

The authors have cited one (ref#59).

Per the recommendation, we have added a section on the combination of immune checkpoint inhibitors and oncolytic virus. We have also added a separate table (Table #5) highlighting some of these combination clinical trials. Both Ribas as well as Chesney studies are m

Minor issues:

1.Page 6, lines 201-204. The authors missed an important fact: This phase III trial with Pexa-vec has been terminated in August of 2019, due to ineffctivenee.

https://www.cancertherapyadvisor.com/home/cancer-topics/general-oncology/phase-3-trial-for-oncolytic-viral-therapy-pexa-vec-in-advanced-liver-cancer-terminated-early/

We have made the correction

2.Page 6. Line 219: there is a typo. It is “Senaca Valley virus”, not “Senava…”.

    • We have made the correction

3.Page 7. Section of Phase II trial. The authors missed a phase II trial with NTX-010 (the Seneca Valley virus). It is Schenk EL et al. J Thorac Oncol. 2020; 15: 110-119.

We have added the description of this study under SVV section

Sincerely,

Aman Chauhan, MD

.

Reviewer 2 Report

The review is clearly written and focus on published clinical trials using oncolytic viruses. The review describes the basics of mechanisms of action and outcomes in clinical trials.

However, the review does not feel "front line" as there are many new concepts of oncolytic viruses, including a plethora of interesting immunostimulatory gene inserts while the authors focus on older published studies describing GM-CSF for example. Hence, the review would benefit from a section discussing what is new and hot in the field including a table with ongoing clinical trials. Such a section should also include a more detailed description of the OV combination with checkpoint blockade antibodies such as the newest data from Tvec, CAVATAC etc.

Author Response

Reviewer #2

The review is clearly written and focus on published clinical trials using oncolytic viruses. The review describes the basics of mechanisms of action and outcomes in clinical trials.

However, the review does not feel "front line" as there are many new concepts of oncolytic viruses, including a plethora of interesting immunostimulatory gene inserts while the authors focus on older published studies describing GM-CSF for example. Hence, the review would benefit from a section discussing what is new and hot in the field including a table with ongoing clinical trials. Such a section should also include a more detailed description of the OV combination with checkpoint blockade antibodies such as the newest data from Tvec, CAVATAC etc.

Thank you for your suggestions. While writing the manuscript, we were tried to focus on the current drug development status of various oncolytic viruses in clinical trials; we deliberately left the mechanistic section lean. However, we completely agree with your observation and have now added important studies leveraging these novel genetic modifications. I hope you find these additions satisfactory.

We also appreciate your suggestions on a dedicated oncolytic virus and immune checkpoint clinical trial section as well as a dedicated table. We have now added both and covered some critical combination studies.

Round 2

Reviewer 2 Report

I agree with the review even if still a bit weak on describing the explosion of new OVs. However, you did a good job to better describe the combination with CPIs in the revised vision that compensate for the my other comment.